# Nanoindentation Creep, Elastic Properties, and Shear Strength Correlated with the Structure of Sn-9Zn-0.5nano-Ag Alloy for Advanced Green Electronics

**Asit Kumar Gain [1] and Liangchi Zhang [2],***

[1]   Laboratory for Precision and Nano Processing Technologies, School of Mechanical and Manufacturing Engineering, University of New South Wales, Sydney, NSW 2052, Australia; a.gain@unsw.edu.au

[2]   Department of Mechanics and Aerospace Engineering, Southern University of Science and Technology, Shenzhen 518055, Guangdong, China

*   Correspondence: zhanglc@sustech.edu.cn; Tel.: +86-0755-8801-8171

**Abstract:** This work investigates the influence of an Ag nanoparticle addition on the microstructure, microhardness, creep, temperature-dependent elastic properties, damping capacity, and shear strength of an environmentally friendly eutectic Sn-9Zn (wt.%) material. A microstructure analysis confirmed that adding Ag nanoparticles significantly altered the morphologies of the Zn-rich phase, which includes the size and shape in the presence of fine spherical-shaped $AgZn_3$ intermetallic compound (IMC) particles in the β-Sn matrix. These fine microstructures positively impact on microhardness, creep, damping capacity, and temperature-dependent elastic properties. Furthermore, in the electronic interconnection on an Au/Ni-plated-Cu pad ball grid array (BGA) substrate, adding Ag nanoparticles generates an additional $AgZn_3$ IMC layer at the top surface of the $AuZn_3$ IMC layer. It also significantly improves the oxidation resistance of Sn-Zn material due to the formation of fine $AgZn_3$ IMC particles. Moreover, the interfacial shear strength value of the Sn-Zn material doped with Ag nanoparticles on the Au/Ni-Cu pad BGA substrate increased about 12% as compared to the reference material after five minutes of reaction in the presence of a fine Zn-rich phase and $AgZn_3$ IMC particles, which acted as second phase dispersion strengthening mechanism. Adding Ag nanoparticles also altered the fracture mode to a typical ductile failure with rough dimpled surfaces of the Sn-Zn material.

**Keywords:** lead-free green alloy; microstructure; creep property; damping capacity; shear strength

## 1. Introduction

In modern electronic packaging systems, electronic interconnection plays an important role in providing mechanical and electrical support to chips and integrated circuits (ICs) on printed circuit boards (PCBs), which directly influences the reliability and performance of electronic devices [1,2]. Until the early eighties, eutectic or near eutectic Sn-Pb-based materials were widely used in microelectronic systems [3–6]. However, their inherent toxicity to human health and the environmental restriction of their applications [7–10] caused them to be replaced with an environmentally friendly material with a similar melting temperature and mechanical properties. Recently, a wide range of environmentally friendly Sn-based materials associated with different weight percentages of metallic particles—for example, Sn-10Sb (271 °C), Sn-58Bi (138 °C) Sn-9Zn (198 °C), Sn-3.5Ag (221 °C), Sn-0.7Cu (227 °C), and Sn-Ag-Cu (217 °C)—were introduced for green electronic devices [11–14]. Among the above-mentioned Sn-based alloys, the cost-effective and better mechanical properties of eutectic Sn-9Zn

material, with a low melting temperature that is close to that of the traditional Sn-37Pb alloy (183 °C), have caused it to be considered one of the best choices for applications in green electronic products. However, its poor oxidation resistance and micro-void formation need to be resolved to expand its applications in modern green electronics [15].

Moreover, the trend towards the miniaturization of electronic packaging systems has increased the current density across interconnections in order to decrease their sizes. For example, the current design of a 100 μm-size electronic interconnection has a current density of $2 \times 10^4$ A/cm$^2$. However, in the near future, the size of this connection will reduce to 1 μm, which can cause the current density to rise to approximately $1 \times 10^7$ to $1 \times 10^8$ A/cm$^2$ [16]. Therefore, conventional materials no longer grantee the reliability of electronic interconnections in modern electronic packaging systems. To develop a highly reliable electronic interconnection material for miniaturized green electronic devices, several research groups have concentrated on introducing an environmentally friendly Sn-based composite material doped with various types of metallic and ceramic—e.g., Al, Ni, Sb, Cu, Ga, Ag, TiO$_2$, CeO$_2$, ZrO$_2$, etc.—particles [17–21]. For example, Chen et al. [22] developed a series of Sn-Zn-based materials doped with Ga particles and reported that adding 0.5% Ga improved the wettability significantly. Further, Chang et al. [23] fabricated a 0.5 wt.% Ag micro-particle-doped Sn-Zn material and evaluated its oxidation and corrosion behavior. Experimental analysis confirmed that adding micron-sized Ag particles improved the oxidation and corrosion resistance of Sn-Zn materials. Furthermore, the electronic interconnect materials generated heat during turning the systems off or on. The overheating may induce the viscous deformation of the interconnection material. Moreover, modern devices are also frequently used in a wide range of frequencies/vibrations, which also creates a new challenge for the performance and longevity of the electronic system [24]. Thus, this is one of the critical factors in understanding the energy dissipation behavior—"so called damping capacity"—of lead-free Sn-based material. Therefore, it is fundamental scientific interest to understand the effect of Ag nanoparticles on the structure, microhardness, creep, shear strength, and temperature dependence of elastic properties, such as the storage modulus, loss modulus, and damping capacity of an environmentally friendly Sn-9Zn-based material.

The present study investigates the influence of an Ag nanoparticle addition on the microstructure and material properties of an environmentally friendly Sn-9Zn-based material for green electronic devices. The aims of this work are (i) to observe the microstructural change in the Sn-Zn-based material doped with Ag nanoparticles; (ii) to understand the microhardness, creep, and temperature dependence of elastic properties, such as the storage modulus, loss modulus, and damping capacity of an Sn-Zn-based material; (iii) to evaluate the oxidation behavior of Sn-Zn-based material under a high temperature (85 °C) and relatively humid (85%) conditions; and (iv) to measure the shear strength of the electronic interconnection of an Au/Ni-plated BGA substrate.

## 2. Materials and Methods

### 2.1. Materials and Microstructure Observation

Bulk Sn-9Zn-based composite electronic interconnect material doped with Ag nanoparticles was manufactured by a casting process. At first, a eutectic Sn-9Zn-based solder paste (Showa Denko JUFFIT-E 9ZSN10M, Shenzhen, China, purity 99.5%) with a particle size about 20–30 μm in diameter was homogeneously mixed with 80–120 nm-diameter Ag particles (Shenzhen Junye Nano Material Co., Shenzhen, China) by a mechanical mixing process. Then, the mixture materials—e.g., Sn-9Zn-0.5Ag—were melted in a ceramic boat at 245 °C in an inert atmosphere for fabricating the cast ingots. Later, the ingots were cut into small pieces and polished for structural investigation and mechanical testing. Further, for making the electronic interconnections on organic solderability preservative (OSP)-Cu and Au/Ni-plated Cu pad ball grid array (BGA) substrates, first the paste mixtures were printed on alumina plates through a stainless steel stencil printing technique and reflowed in a BTU VIP-100N oven (Pyramax 100N, North Billerica, MA 01862, USA) for preparing

the solder balls with 760 μm diameters. Later, the solder ball was attached on the OSP-Cu and Au/Ni-plated Cu (BGA) substrate and reflowed at 245 °C using a BTU VIP-100N oven in an inert atmosphere. After preparing the electronic interconnections, one set of specimens was put in an ESPEC-PL-2FP temperature and relative humidity (RH) chamber with a constant temperature (85 °C) and humidity (85%). Later, for understanding the influence of Ag nanoparticles on the structure of Sn-9Zn-based material, the cast ingot and solder joints were cross-sectioned and polished by a metallographic technique and their structure was observed using a scanning electron microscope (Hitachi S3400, Hitachi Science Systems Ltd. Tokyo, Japan) with backscattered electron mode.

### 2.2. Mechanical Properties Measurements

The microhardness of Sn-9Zn material and material doped with Ag nanoparticles was measured by a DuraScan hardness testing machine (EMCO-TEST Prüfmaschinen GmbH, Kuchl, Austria) with a mapping system of about 700 points with a 1N load and 10s dwelling. Later, the temperature-dependent elastic properties—for example, the storage modulus, loss modulus, and damping capacity of Sn-9Zn-based material—were measured using a dynamic mechanical analyzer (DMA, TA 2980, TA Instruments, Inc. New Castle, DE, USA) at different operating temperatures with a constant frequency (1 Hz). The dimension of the temperature-dependent elastic properties measurement specimens was about 15.5 mm × 6.7 mm × 1.91 mm. Later, the creep performance of Sn-9Zn-based material was carried out by a nano-indentation (Hysiron TI-950 Tribo Indenter, Bruker, Minneapolis, MN, USA). During this measurement, a constant load of 10,000 μN was applied for dwelling 30 s with a series pattern of thirty-six points. Further, the shear strength of electronic interconnections on Au/Ni-plated Cu pad was measured by ball shear test (PTR-1000, Rhesca Co., Ltd., Tokyo, Japan) with a constant speed of 500 μm/s of twenty-five joints. After measuring the shear strength of electronic interconnections, the fracture surface morphology was characterized by the SEM technique.

## 3. Results and Discussions

### 3.1. Microstructure of Bulk Sn-Zn-Based Material

The microstructure of the reference Sn-9Zn material and material doped with Ag nanoparticles was observed by SEM, and the results are presented in Figure 1. From the structural analysis, it was revealed that the needle-shape Zn-rich phase was evenly distributed in the β-Sn matrix in both types of materials. However, interesting phenomena were observed after adding Ag nanoparticles, and it was found that the aspect ratio of the Zn-rich phase was changed significantly and appeared to have a fine structure. The size of this Zn-rich phase in the reference material was in the range of 9–18 μm in length, whereas its value reduced to 3–10 μm in length in the material doped with Ag nanoparticles. Furthermore, very fine spherical-shaped $Ag_3Zn$ IMC particles were clearly detected in the Sn-Zn material doped with Ag nanoparticles. According to an EDS analysis, it was confirmed that the spherical-shaped $Ag_3Zn$ IMC particles consist of 79.8 at.% Zn and 20.2 at.% Ag elements. The main reason for evaluating the fine microstructure in this material was because the addition of Ag nanoparticles may change the chemical affinity and diffusivity.

### 3.2. Microhadness and Creep of Sn-Zn-Based Material

The microhardness value of a material relies on the movement of dislocation as well as the growth nature and configuration of the grain structure. The processing parameters are more effective for developing the microstructure of the electronic interconnect material than their chemical composition. Therefore, the material properties—e.g., the microhardness value of the electronic interconnect material—depends on the microstructure, fabricating temperature, and its composition. Figure 2 shows the microhardness of (a) the reference material and (b) the Sn-Zn material doped with Ag particles and their 3D mapping images. According to the microhardness measurement, it was clearly indicated that the value of microhardness of the reference Sn-Zn material was about 13.4 ± 0.982 HV, while its

value was indicated to be about 17.7 ± 0.917 HV in the material doped with Ag nanoparticles. Overall, the microhardness value of the Sn-Zn material doped with Ag nanoparticles increased approximately 32% as compared to the reference material. The main reason for the higher microhardness of the Ag nanoparticle-doped material was the presence of the spherical-shaped $Ag_3Zn$ IMC particles and fine Zn-rich phases in the matrix. The fine microstructure contributes to the dispersion-strengthening mechanism in the β-Sn matrix.

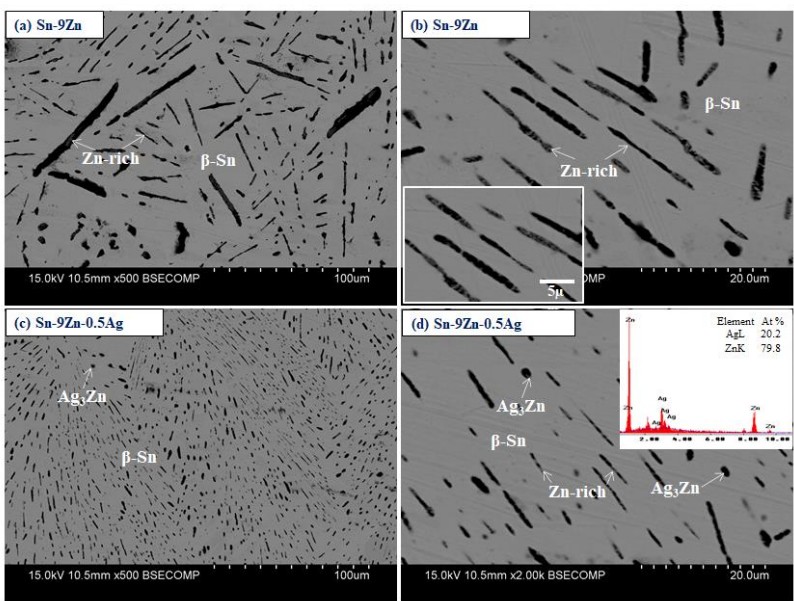

**Figure 1.** SEM images of (**a**,**b**) reference Sn-Zn material and (**c**,**d**) Sn-Zn material doped with Ag nanoparticles.

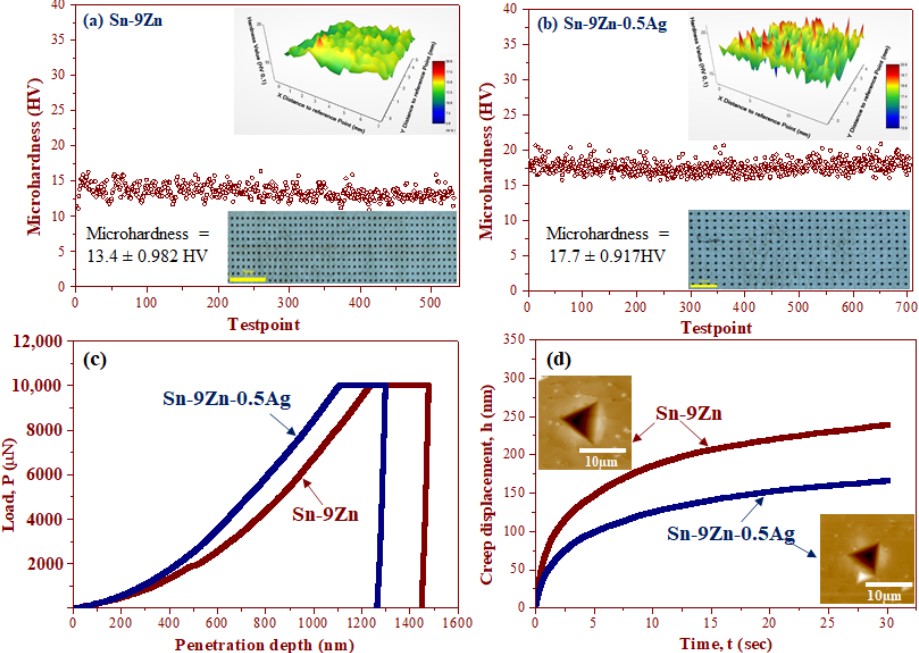

**Figure 2.** Microhardness of (**a**) the reference material and (**b**) the Sn-Zn material doped with Ag particles. (**c**) Nanoindentation load–penetration depth curves and (**d**) variations in penetration depth at maximum load during the holding time for Sn-Zn-based material.

Generally, the electronic packaging systems are often used in harsh operating environments and also generated heat during their self-operation, such as the switching on/off of the devices. This heat generation can influence the creep of the interconnect material. Furthermore, it is well known that the environmentally friendly Sn-based materials exceed half of the homologous temperature (TH = $T/T_{mp}$), even when operated at room temperature. Thus, understanding the creep of lead-free Sn-Zn material is one of the most crucial factors in applying this material in miniaturized green electronic devices. The creep of the reference Sn-Zn material and material doped with Ag nanoparticles was studied by nanoindentation. For the creep evaluation of Sn-based materials, a similar indentation load (10,000 µN) was applied in the earlier literature [25]. From the load versus penetration depth and creep displacement at maximum load, as presented in Figure 2c,d, the penetration depth of the reference material was higher than that of the Sn-Zn material doped with Ag nanoparticles. This analysis indicated that adding Ag particles enhanced the creep resistance of the Sn-Zn alloy. This result was further confirmed by AFM images of the permanent indentation marks, as can be seen in Figure 2d.

### 3.3. Temperature Dependence of the Elastic Properties of Sn-Zn-Based Material

Electronic packaging materials are commonly used at various service environments that are mainly integrated with different types of thermal and mechanical shocks. The variation in thermo-mechanical loading can cause the electronic interconnections to fail. This is more severe in downscaled electronic interconnections, and thus creates a new challenge to the reliability and longevity of the miniaturized green electronics. Therefore, understanding the temperature-dependent elastic properties of environmentally friendly Sn-based material plays a key role for designing an advanced green electronic device. Figure 3a shows the storage modulus and loss modulus of Sn-Zn-based material as a function of the operating temperature, measured at a constant frequency of 1 Hz. Overall, the storage modulus of Sn-Zn material doped with Ag nanoparticles displayed a higher value as compared to the reference material. However, their vales were decreased when increasing the operating temperature. Consequently, its loss modulus displayed a lower value as a function of the operating temperature. This measurement indicated that the Ag nanoparticle-doped Sn-Zn material has a better thermo-mechanical stability than that of the reference material. Figure 3b shows the temperature dependence of the damping capacity of the reference material and material doped with Ag nanoparticles measured at a constant frequency (1 Hz). Usually, the damping capacity of a material can be estimated by the ratio of the loss modulus to the storage modulus. Sutou et al. described in detail its physical importance in his original research [26]. From this analysis, it was well recognized that the damping capacity values were gradually raised with the operating temperature in both specimens; see Figure 3b. Overall, the damping resistance of the Ag nanoparticle-doped Sn-Zn material was better compared to that of the reference material. A plausible explanation for the better damping resistance of the Ag nanoparticle-doped Sn-Zn material is that the fine microstructure significantly reduced the values of the loss modulus compared to the reference material. Further, Figure 3c,d show the strain amplitude dependence of the damping capacity of Figure 3c the reference material and Figure 3d the material doped with Ag nanoparticles at various operating temperatures measured at a constant frequency (1 Hz). At low strain, the damping capacity of all the specimens appeared with low values, and their values were gradually increased with strain. Furthermore, at a high operating temperature, the damping resistance of all the specimens was reduced drastically due to the increase in their loss modulus, as well as the decrease in their storage modulus. Overall, the damping resistance of the Ag nanoparticle-doped Sn-Zn material has a higher value than that of the reference material.

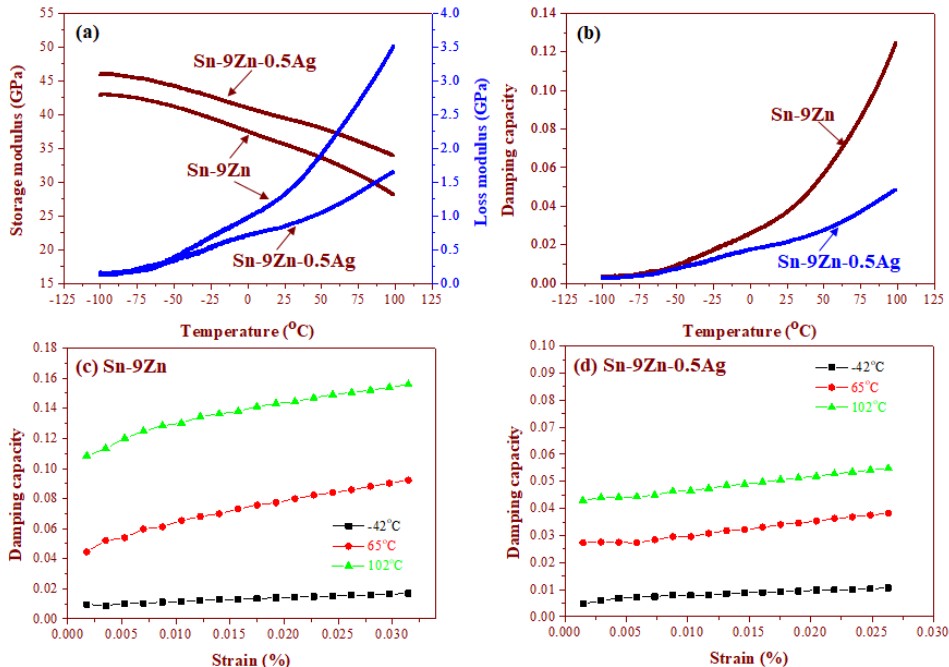

**Figure 3.** (**a**) Temperature-dependent storage modulus and loss modulus of the Sn-Zn-based material and (**b**) its damping capacity. Strain amplitude-dependent damping capacity of (**c**) the reference material and (**d**) the Sn-Zn material doped with Ag nanoparticles measured at various temperatures with a constant frequency (1 Hz).

### 3.4. Interfacial Microstructure of Sn-Zn-Based Material on OSP-Cu and Au/Ni BGA Pads

The electronic interconnection on the OSP-Cu pad was prepared according to the stencil printing and reflow process according to the schematic diagram in Figure 4a, and their wettability was characterized. According to the wettability analysis, it was obvious that adding Ag nanoparticles increased the spreading area by about 13% as compared to the reference material. Consequently, its contact angle value decreased to 10.8%. This finding confirmed that adding Ag nanoparticles improved the wettability of the Sn-Zn-based material, which can positively impact on the mechanical reliability of the electronic interconnections. In earlier literature, Kripesh et al. developed an empirical relationship between the wettability and the contact angle. They mentioned a "very good" wettability when the contact angle was $0° < \theta < 20°$. For the value $20° < \theta < 40°$, it was considered as "good and acceptable". The wettability was "bad" when $\theta > 40°$ [27]. The contact angle of the Sn-Zn material doped with Ag nanoparticles was about 28.9°, which presented a good wettability. Furthermore, according to the interfacial microstructure analysis in the Sn-Zn/OSP-Cu system (Figure 4b,c), a scallop-shaped binary $Cu_5Zn_8$ IMC grew at the substrate surface and the elemental analysis confirmed that it consists of 56.8 at.% Zn and 43.2 at.% Cu elements. However, after adding Ag nanoparticles, an additional $AgZn_3$ IMC layer was found to be adhered to the top surface of the $Cu_5Zn_8$ IMC layer, as shown in Figure 4d,e, and this consists of 79.3 at.% Zn and 20.7 at.% Ag elements. Song et al. reported that $AgZn_3$ IMC grew at the Cu substrate surface due to a subsequent peritectic reaction of $L + \gamma\text{-}Ag_5Zn_8 \rightarrow \varepsilon\text{-}AgZn_3$ [28].

Figure 5 shows interfacial SEM images of Figure 5a,b the reference material and Figure 5c,d Sn-Zn material doped with Ag nanoparticles on Au/Ni plated BGA pad as a function of reaction time of Figure 5a,c 5 min and Figure 5b,d 30 min. From a structure analysis of the reference material, a scallop-shaped $AuZn_3$ IMC layer was grown at the substrate surface as a result of the fast reaction of the Au and Zn elements. However, when increasing the reaction time, this IMC layer was detached from the substrate surface, as shown in Figure 5b. In addition, some Kirkendall voids, as indicated with the arrow heads in Figure 5a, were also detected at their interfaces. On the other hand, in the

interconnections made by the Ag nanoparticle-doped material, an additional dark contrast AgZn₃ IMC layer was found to be adhered to the top surface of the AuZn₃ IMC layer, with an absence of Kirkendall voids, as shown in Figure 5c,d. The absence of Kirkendall voids in the electronic interconnection made by the Ag nanoparticle-doped Sn-Zn material is because adding nanoparticles changed the diffusion process, which may have blocked the formation of Kirkendall voids.

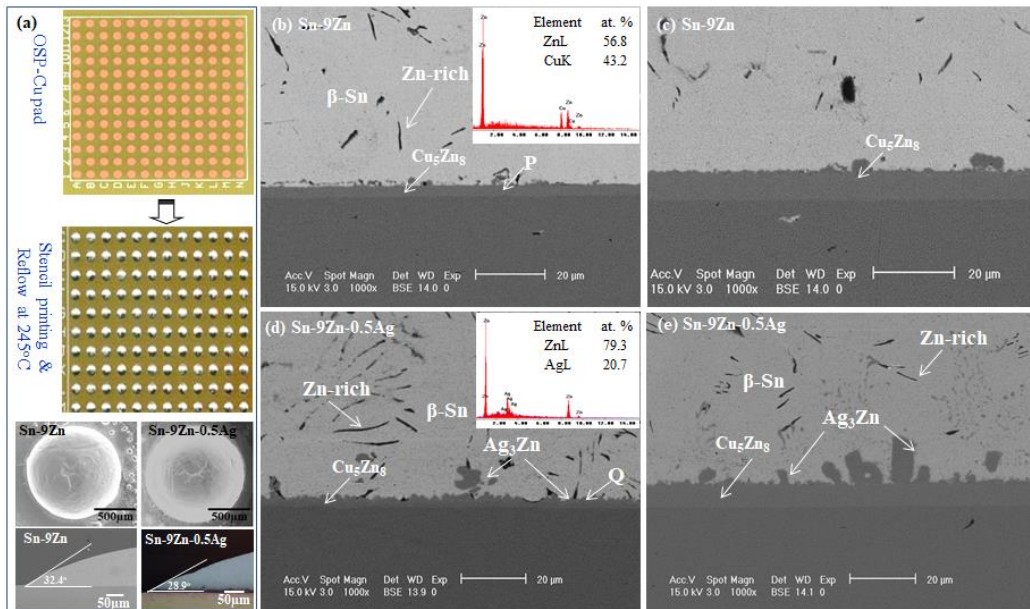

**Figure 4.** (**a**) Manufacturing process of the electronic interconnection on the Organic Solderability preservative (OSP)-Cu pad. Interfacial SEM images of (**b**,**c**) the reference material and (**d**,**e**) the Sn-Zn material doped with Ag nanoparticles as a function of the reaction times of (**b**,**d**) 5 min and (**c**,**e**) 30 min.

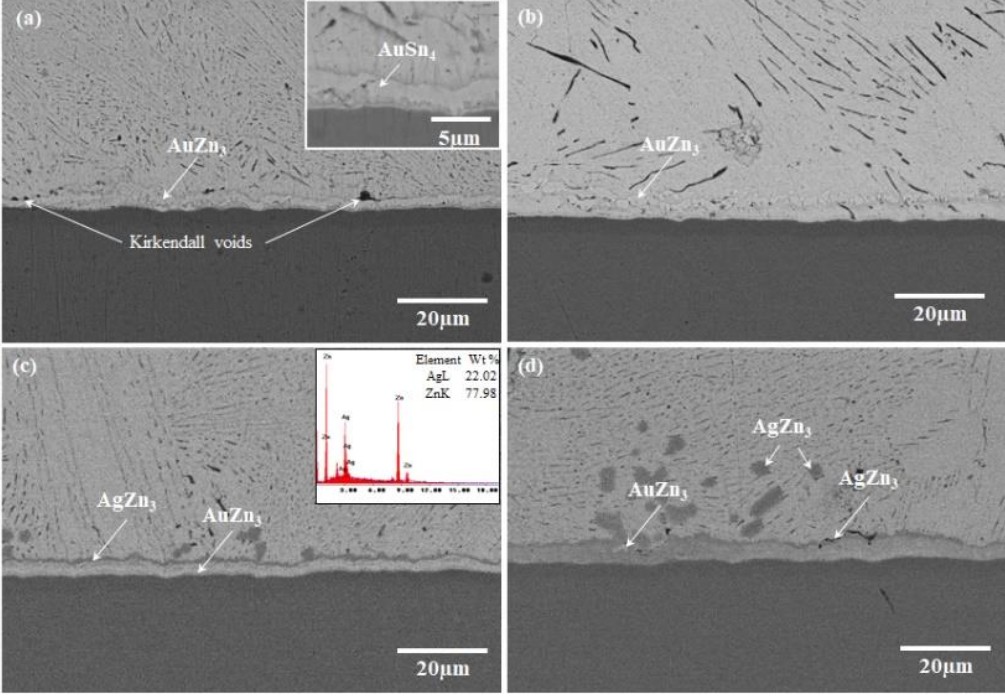

**Figure 5.** Interfacial SEM images of (**a**,**b**) the reference material and (**c**,**d**) Sn-Zn material doped with Ag particles on Au/Ni plated BGA pad as a function of reaction time of (**a**,**c**) 5 min and (**b**,**d**) 30 min.

### 3.5. Temperature and Humidity Effect on Sn-Zn-Based Material

An environmentally friendly eutectic Sn-9Zn-based material has been recognized as a suitable alloy to eliminate toxic Pb-based material in advanced green electronic devices. However, their poor oxidation resistance in harsh service environments—e.g., under high temperature and high relative humidity (RH)—minimizes their applications in advanced devices. Thus, for understanding the oxidation behavior, the Sn-9Zn-based material was exposed to a high-temperature (85 °C) and high-relative humidity (RH) (85%) environment, and the outcomes are shown in Figure 6. From this analysis, it is obvious that the Zn-rich phase was oxidized from the top surface and penetrated in the solder matrix. However, the more interesting phenomena observed were that the oxidation behavior of the reference material was faster than that of the Sn-Zn material doped with Ag nanoparticles. The plausible explanation for the apparent high oxidation resistance of the Ag nanoparticle-doped Sn-Zn-based material was because adding the Ag nanoparticles grew fine $AgZn_3$ IMC particles and decreased the Zn-rich phase in the β-Sn matrix. Furthermore, the growth of these IMC particles inhibited the penetration of oxygen and vapor. In an earlier study, Lee et al. [29] investigated the oxidation behavior of Sn-Zn-based material and confirmed that the presence of the Zn-rich phase in the eutectic Sn-Zn alloy reduced or fixed forming IMCs, which enhanced the oxidation resistance of the Sn-Zn alloy. In the reference material after exposure at 85 °C/85% RH for 1000 h as shown in Figure 6b, the solder matrix was fully oxidized and grew some microcracks, as indicated with the arrowhead-inserted image in Figure 6b. The reason for the growing microcracks in the matrix is that the oxidation of the Zn phase to ZnO was accompanied by a volume expansion. As the oxidation progresses, the stress induced by the volume expansion of the ZnO phase, especially along the β-Sn grain boundaries, causes cracks in the matrix.

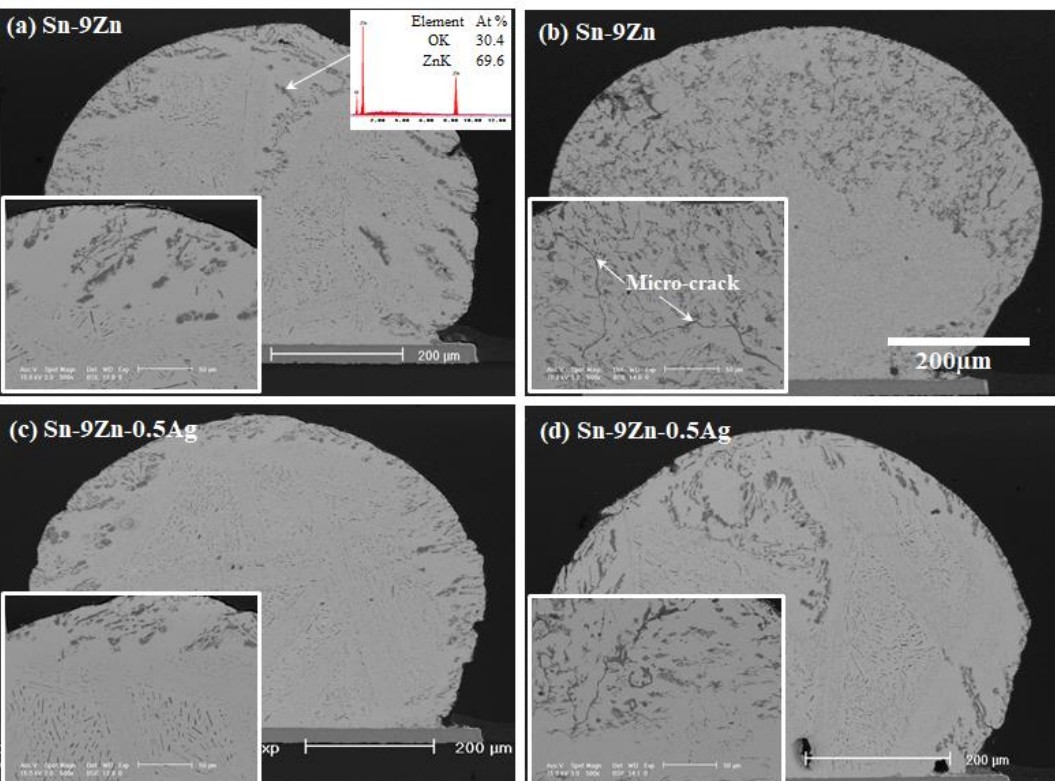

**Figure 6.** SEM images of (**a**,**b**) the reference material and (**c**,**d**) the Sn-Zn material doped with Ag particles after exposure to a high-temperature and high-humidity environment as a function of the exposure time of (**a**,**c**) 100 h and (**b**,**d**) 1000 h.

*3.6. Shear Strength of Sn-Zn-Based Material on Au/Ni-Plated BGA Pad*

Figure 7a presents the shear strength of the reference material and the material doped with Ag nanoparticles as a function of the reaction time. From this measurement, it was obvious that the electronic interconnection made by the Sn-Zn material doped with Ag nanoparticles exhibited a higher strength as compared to the electronic interconnection made by the reference material, and its value was increased about 12%. The shear strength value of the Sn-Zn-based material was reduced when increasing the reaction time. The main reason for the higher shear strength of the Ag-doped material was the fine microstructure as well as the formation of fine IMC particles in the solder matrix. The evenly distributed IMC particles and fine Zn-rich phase acted as a second-phase dispersion-strengthening mechanism in the β-Sn matrix. After investigating the shear strength, the fracture surface morphologies were observed by SEM, and the results are presented in Figure 7a,b. From these fractographs, the reference material fracture surface appeared to have a brittle fracture with a smooth surface (Figure 7b). However, after adding the Ag nanoparticles, the fracture mode was changed and appeared to be a ductile fracture with dimpled surfaces (Figure 7c) due to the formation of fine $AgZn_3$ IMC particles.

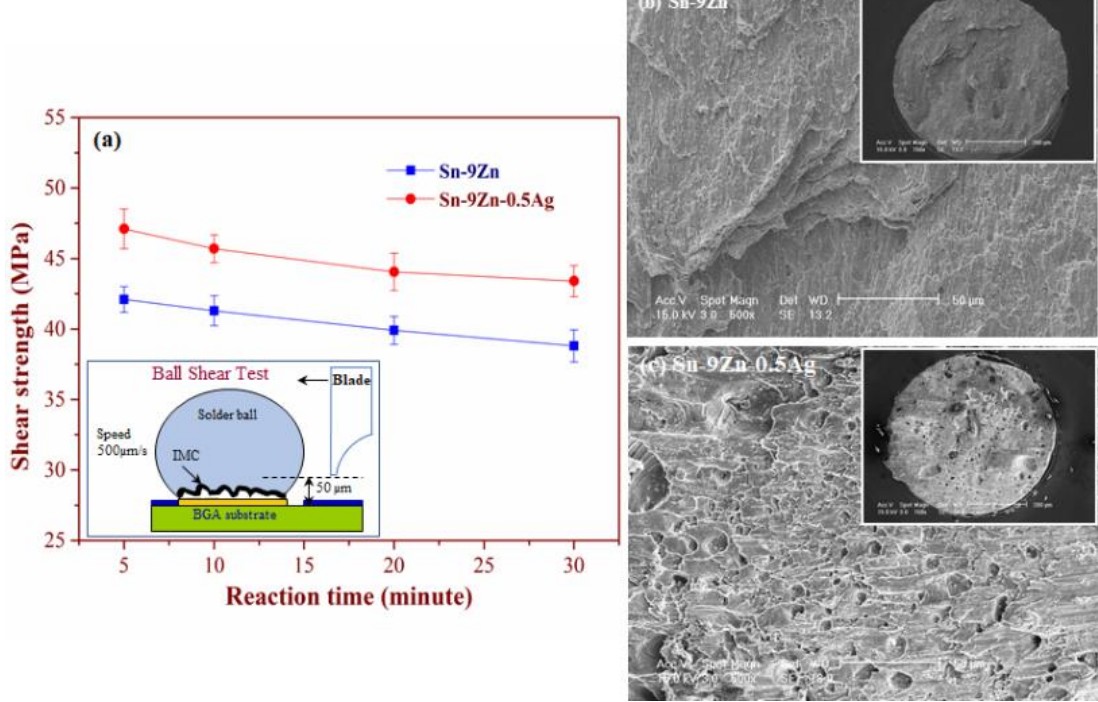

**Figure 7.** (**a**) Shear strength of the Sn-Zn-based material. SEM fracture surfaces of (**b**) the reference material and (**c**) the Sn-Zn material doped with Ag nanoparticles on an Au/Ni-plated Ball Grid Array (BGA) pad.

## 4. Conclusions

The present study has comprehensively investigated the effect of Ag nanoparticles on the microstructure, microhardness, creep, temperature-dependent elastic properties, damping capacity, and shear strength of Sn-9Zn material. The experimental results give us the following understanding:

1.　The structural-property analysis revealed that nanosized-Ag particles refined the microstructure in the presence of fine spherical-shaped $AgZn_3$ IMC particles in the β-Sn matrix. These fine microstructures increased the microhardness, creep, and damping resistance as well as the storage modulus as compared to the reference Sn-9Zn material due to the second phase dispersion-strengthening mechanism.

2. In an interfacial microstructure analysis of the OSP-Cu pad/Sn-Zn-0.5Ag system, an additional AgZn$_3$ IMC layer was found to be adhered to the top surface of the Cu$_5$Zn$_8$ IMC layer. On the other hand, in the Au/Ni-plated-Cu pad BGA pad/Sn-Zn-0.5Ag system, sandwich-type AgZn$_3$ and AuZn$_3$ IMC layers had grown at the substrate surface. Furthermore, the addition of Ag nanoparticles also enhanced the oxidation resistance of the Sn-Zn material due to the formation of fine AgZn$_3$ IMC particles.

3. The addition of Ag nanoparticles increased the shear strength value of the Sn-Zn material in the presence of a fine Zn-rich phase and AgZn$_3$ IMC particles by about 12% with changing the fracture mode to a typical ductile failure with rough dimpled surfaces.

**Author Contributions:** L.Z. supervised and directed this project. A.K.G. performed the experiments and prepared the manuscript. All authors have read and agreed to the published version of the manuscript.

**Funding:** This research was funded by UNSW, grant number DP190102959.

**Acknowledgments:** Authors thank to Tit Wah Chan, Department of Physics and Materials Science, CityU Hong Kong, for helping in evaluating the damping capacity.

**Conflicts of Interest:** The authors declare no conflict of interest.

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
