# Peer review of "Nanoindentation Creep, Elastic Properties, and Shear Strength Correlated with the Structure of Sn-9Zn-0.5nano-Ag Alloy for Advanced Green Electronics"

_metals, doi:10.3390/met10091137_

Round 1

Reviewer 1 Report

The authors showing interesting findings of Sn-9Zn-0.5nano-Ag alloy for advanced green electronics. The manuscript is well organized and conclusions are significant for the field of green soldering technology.

However, there are some commnets that needs to be considered:

Introduciton section:

Page 2, Line 64.: "Thus, it is one of the critical factors to understand the energy dissipation behavior “so called damping capacity” of lead-free Sn-based material under temperature and vibration." .... Under what temperature? Higher or room temperature? Please specify. 

Materials and Methods section:

Page 3, Line 82.: The authors should mention also the volume of Ag particles mixed with the Sn-9Zn based solder paste. What was the final composition of new Sn-9Zn-xAg solder? In the results and discussion section you mentioned Sn-9Zn-0.5Ag. For better understanding, You should also mention it in this section.

Results and discussion section:

Figure 1: How did you identified the Ag3Zn IMC? Some XRD or EDX analysis? 

Author Response

Reviewer 1

The authors showing interesting findings of Sn-9Zn-0.5nano-Ag alloy for advanced green electronics. The manuscript is well organized and conclusions are significant for the field of green soldering technology.

However, there are some comments that needs to be considered:

Introduciton section:

Page 2, Line 64.: "Thus, it is one of the critical factors to understand the energy dissipation behavior “so called damping capacity” of lead-free Sn-based material under temperature and vibration." .... Under what temperature? Higher or room temperature? Please specify. 

Author: We modified the sentence.

Thus, it is one of the critical factors to understand the energy dissipation behavior “so called damping capacity” of lead-free Sn-based material.

Materials and Methods section:

Page 3, Line 82.: The authors should mention also the volume of Ag particles mixed with the Sn-9Zn based solder paste. What was the final composition of new Sn-9Zn-xAg solder? In the results and discussion section you mentioned Sn-9Zn-0.5Ag. For better understanding, You should also mention it in this section.

Author: We mentioned the composition in Materials and Methods section [Page 3, Line 83].

Results and discussion section:

Figure 1: How did you identified the Ag3Zn IMC? Some XRD or EDX analysis? 

Author: We identified the Ag3Zn IMC by EDS analysis data and added description.

According to EDS analysis, it was confirmed that the spherical-shaped Ag3Zn IMC particles consists of 79.8 at.% Zn and 20.2 at.% Ag elements.

Reviewer 2 Report

Comments on the reviewed paper:

  1. On page 2, line 18, the Authors indicate that the size of Zn-rich phase particles was 15-18 µm, whereas Fig. 1a shows much larger precipitates. Is the presented result an average value or a range of sizes of these precipitates?
  2. On what basis do the Authors clearly indicate the occurrence of these specific phases as shown in Fig. 1, and similarly in Figs. 4 and 5?
  3. Line 131-33, the sentence ”Microhardness mapping...” is a repetition of information included in the “Materials and Methods”.
  4. Can the value of standard deviation be given for the obtained results of microhardness?
  5. According to the Authors, what the good wettability of Sn-Zn-Ag alloy may result from?

Author Response

1. On page 2, line 18, the Authors indicate that the size of Zn-rich phase particles was 15-18 µm, whereas Fig. 1a shows much larger precipitates. Is the presented result an average value or a range of sizes of these precipitates?

Answer 1: The presented result is a range of sizes of these precipitates?

The size of this Zn-rich phase in the reference material was in a range of 9-18 µm in length whereas its value reduced to 3-10 µm in length in the material doped with Ag nanoparticles.

2. On what basis do the Authors clearly indicate the occurrence of these specific phases as shown in Fig. 1, and similarly in Figs. 4 and 5?

Author: We identified the Ag3Zn IMC by EDS analysis data and added description.

According to EDS analysis it was confirmed that the spherical-shaped Ag3Zn IMC particles consists of 79.8 at.% Zn and 20.2 at.% Ag elements.

3. Line 131-33, the sentence “Microhardness mapping...” is a repetition of information included in the “Materials and Methods”.

Author: We modified the sentence

“Microhardness mapping of Sn-Zn-based material was carried out using the DuraScan hardness tester and their hardness values and 3D mapping images were presented in Figure 2.”

Figure 2 shows microhardness of (a) the reference material and (b) Sn-Zn material doped with Ag particles and their 3D mapping images.

4. Can the value of standard deviation be given for the obtained results of microhardness?

Author: We added the value of standard deviation in Figure 2 and modified the sentence

According to the microhardness measurement, it was clearly indicated that the value of microhardness of the reference Sn-Zn material was about 13.4 ± 0.982 HV while its value indicated about 17.7 ± 0.917 HV in the material doped with Ag nanoparticles.

5. According to the Authors, what the good wettability of Sn-Zn-Ag alloy may result from?

Author: We described it in page 7 line 202-206.

Kripesh et al. developed an empirical relationship between the wettability and contact angle. They mentioned as a “very good” wettability when the contact angle is 0o<?<20o. For the value 20o<?<40o, it is considered as “good and acceptable”. The wettability is “bad” when ?>40o [27]. The contact angle of Sn-Zn material doped with Ag nanoparticle was about 28.9o which presented an acceptable and good wettability.

Reviewer 3 Report

The paper investigates the influence of nanoparticals addition on the properties of Sn-9Zn based materials. The paper is well organized and easily readable. I don't see any repetition and the method seems rigorous. The authors have published several papers on this topic and consequently the originality is reduced.

Here are my comments:

  • Page 2 - line 60: Please revise the sentence about the creep - The overheating may induce viscous deformation of the interconnection material
  • Suggestion: About the creep responce the comparison was performed comparing exclusively the maximum penetration depth after 30 seconds. It's should be interesting compares the creep rates too.

Author Response

The paper investigates the influence of nanoparticles addition on the properties of Sn-9Zn based materials. The paper is well organized and easily readable. I don't see any repetition and the method seems rigorous. The authors have published several papers on this topic and consequently the originality is reduced.

Here are my comments:

Page 2 - line 60: Please revise the sentence about the creep - The overheating may induce viscous deformation of the interconnection material

Author: We modified the sentence.

The overheating may induce viscous deformation of the interconnection material.